# Pneumonia-Related Hospitalizations among the Elderly: A Retrospective Study in Northeast Italy

**DOI:** 10.3390/diseases12100254

**Published:** 2024-10-15

**Authors:** Silvia Cocchio, Claudia Cozzolino, Patrizia Furlan, Andrea Cozza, Michele Tonon, Francesca Russo, Mario Saia, Vincenzo Baldo

**Affiliations:** 1Department of Cardiac, Thoracic, Vascular Sciences and Public Health, University of Padua, 35128 Padua, Italy; silvia.cocchio@unipd.it (S.C.); claudia.cozzolino@phd.unipd.it (C.C.); patrizia.furlan@unipd.it (P.F.); andrea.cozza@studenti.unipd.it (A.C.); 2Regional Directorate of Prevention, Food Safety, Veterinary, Public Health—Veneto Region, 30123 Venice, Italy; michele.tonon@regione.veneto.it (M.T.); francesca.russo@regione.veneto.it (F.R.); 3Azienda Zero of Veneto Region, 35131 Padua, Italy; mario.saia@azero.veneto.it

**Keywords:** pneumonia, *Streptococcus pneumoniae*, elderly, hospital discharge records, vaccination

## Abstract

**Background**: In both the elderly and children, pneumonia remains one of the leading causes of hospitalization. This study aimed to assess the impact of pneumonia-related hospitalizations in the population over 65 years of age in the Veneto Region. **Methods**: This retrospective study analyzed hospital discharge records for patients aged 65 and older who resided in the Veneto Region and had a diagnosis of pneumonia from 2007 to 2023. The hospitalizations were identified using specific ICD-9-CM codes for pneumonia as a discharge diagnosis. Hospitalization rates, mortality rates, the prevalence of complications and comorbidities, the length of stay, and associated costs were calculated by age and year. **Results**: From 2007 to 2023, there were 139,201 hospitalizations for pneumonia. Emergency admissions accounted for 92.1% of these cases, and only 2.0% had a specific diagnosis of pneumococcal pneumonia. The median length of stay was 10 days, and the median diagnosis-related group (DRG) tariff per hospitalization was EUR 3307. Excluding the pandemic years, the hospitalization rates remained stable at approximately 850 cases per 100,000 inhabitants before 2019. After 2022, the rates started to increase again. Overall, in the investigated period, the results showed a negative trend (average Annual Percentage Change (AAPC) of −1.931, *p* < 0.0001). However, when only considering the pre-pandemic years, the trend was stable, while a decline was observed starting in 2020 (AAPC of −19.697, *p* = 0.001). The overall discharge mortality rates ranged from 13% to 19.3% but were significantly higher in those over 85 years of age (20.6% compared with 6.5% and 12.0% in the 65–74 and 75–84 age groups, respectively). **Conclusions**: This study highlights the substantial burden of pneumonia in individuals over 65 years of age, showing the impacts on public health.

## 1. Introduction

Pneumonia, an acute respiratory infection primarily caused by bacteria or viruses, ranges in severity from mild to potentially life-threatening across all age groups, particularly affecting infants and the elderly. The majority of pneumonia cases are community-acquired pneumonia (CAP). The estimated global incidence of CAP varies between 1.5 and 14 cases per 1000 person-years and is influenced by geography, seasonality, and population demographics [1]. Approximately 50% of cases require hospitalization [1].

*Streptococcus pneumoniae* and *Haemophilus influenzae* remain the predominant causative agents worldwide [2,3]. Viral infections are less common (6.8%) [1], with rhinovirus, influenza, and, more recently, severe acute respiratory syndrome coronavirus 2 (SARS-CoV-2) increasingly being identified [3]. However, the microbiological profile and antibiotic resistance significantly differ between CAP and hospital-acquired pneumonia (HAP) cases, and higher rates of *Pseudomonas aeruginosa* and methicillin-resistant *Staphylococcus aureus* (MRSA) infections have been reported [4].

Pneumonia ranks as the eighth leading cause of death globally and is the primary infectious cause of death [2,5]. The mortality rate has reached 0.7 per 1000 persons per year [1]. Apart from children and individuals with underlying health conditions, adults aged 65 and older are particularly vulnerable to this respiratory illness, particularly in high-income countries or aging populations [1,2,6].

Pneumonia in older individuals may present atypically, lacking symptoms, such as fever or productive cough, which can complicate diagnoses. Such elder individuals typically require more intensive treatment compared with younger patients and are at greater risk of developing complications [7]. Despite significant advancements in the diagnosis, treatment, and prevention of pneumonia, the relatively high mortality rates among older adults persist and have reached up to 20% [8,9]. Elderly patients who are bedridden and fed through tubes face significantly higher risks [9].

Elderly individuals diagnosed with pneumonia not only have worse short-term prognoses but also have poor long-term and functional outcomes [8]. Survival of this disease is associated with increased dependence, and the mortality rate is high in the following months and additionally increases to up to 33.6% in the subsequent year [8,10].

Pneumococcal vaccination is the best strategy for reducing the effects of *Streptococcus pneumoniae* infection. The vaccines available to date have played a decisive role in preventing community-acquired pneumococcal pneumonia, and at the same time, they have reduced the risk of invasive pneumococcal disease, severe complications from pneumococcal infection, and hospitalization [11]. It has also been found that in the elderly population, pneumococcal vaccination can reduce the mortality associated with community-acquired pneumococcal pneumonia [11].

Two types of pneumococcal vaccine are available today: the conjugate vaccine (PCV) and the polysaccharide vaccine (PPSV) [11,12,13]. The conjugate vaccine is prepared with different serotypes of *S. pneumoniae* (7, 13, 15, or 20 serotypes) [11,12,13]. Recently (in June 2024), the Food and Drug Administration approved the introduction of a 21-valent conjugate vaccine for use in adults [14].

The 23-valent polysaccharide vaccine contains 23 capsular polysaccharides of *S. pneumoniae* serotypes [12,13]. It is most effective in protecting against invasive pneumococcal disease in individuals up to 75 years of age [13].

Vaccination against *S. pneumoniae* is also of paramount importance in view of the resistance profiles that pneumococcus demonstrates today. Vaccine coverage, by offering protection against pneumococcal disease, reduces the need for antibiotics and indirectly abates the development of resistance [15].

Although results regarding the PCV’s efficacy in children have been reported [16,17,18], the evidence in the elderly is promising but requires further study to understand its scope. Epidemiologic estimates of pneumonia, particularly pneumococcal pneumonia, are essential for such evaluations and for determining the cost-effectiveness of vaccination campaigns in adults. This study aimed to provide a population-based overview of the burden of bacterial pneumonia in individuals aged 65 and older in terms of the need for inpatient hospitalization. Hospitalization rates for pneumonia and co-diagnosed invasive forms, such as meningitis, septicemia, and empyema, were estimated, along with associated direct costs and case fatality rates. Additionally, a trend analysis was conducted on these health outcomes to assess the evolution of pneumonia epidemiology over recent decades.

## 2. Materials and Methods

### 2.1. Study Population and Data Source

Within this population-based study, we carried out a retrospective examination by utilizing hospital discharge records (HDRs) from the Veneto Region. Situated in northeast Italy, Veneto stands as one of the most affluent regions in the nation, boasting a Gross Domestic Product (GDP) of USD 234,995 million in 2023 [19]. In 2022, Veneto was identified as the 13th oldest region in Italy. This region has an age index of 195.1, with a population density of 264.3 individuals per square kilometer. The average population size is 4.8 million, with a mean age of 46.1 years. Additionally, 50.9% of the population is female [20]. The Italian National Health System is a publicly funded system primarily financed through general taxation and organized on a regional basis [21].

According to regional decree no. 118, dated 23 December 2016 [22], each HDR contains one primary discharge diagnosis (or first-listed diagnosis) and up to five secondary diagnoses, based on diagnostic codes. Diagnoses are encoded using the International Classification of Diseases Clinical Modification (ICD-9-CM) system. HDRs also include information about hospitalization regimen (ordinary or day or week surgery admissions), type of admission (scheduled, urgent, outpatient compulsory treatment, or nonurgent delivery), medical or surgical procedures undergone (encoded in ICD-9-PCS), and other administrative data (such as date of admission, transfer, discharge, facility code, etc.) that are essential for calculating expenses. Sociodemographic fields, such as education level and marital status, were recently added, but they were rarely recorded in practice and were, therefore, not used in this study.

### 2.2. Inclusion and Exclusion Criteria

In a retrospective observational framework, we included all HDRs associated with individuals aged 65 and older, generated by inpatient facilities in the Veneto Region from 2007 to 2023, regardless of the type of admission or hospitalization regimen. This study considered all inpatient facilities in the region, both public and private. Cases related to bacterial pneumonia were identified using the discharge diagnosis codes reported in the HDRs (encoded in ICD-9-CM), as described in a previous study [16]. We selected all hospitalizations with a first-listed diagnosis of pneumonia or a first-listed diagnosis of meningitis, septicemia, or empyema associated with a secondary diagnosis of pneumonia (see Table 1). Specifically, we decided not to include pneumonias of mycobacterial, viral, fungal, and parasitic origin in this study. Instead, we decided to consider and include pneumonias classified as "unspecified" in the analysis. This choice was dictated by the assumption, as reported in the literature, that the most frequent cause of pneumonia in adults is bacterial infection [23]. HDRs associated with individuals who were not residing in the Veneto Region were excluded. To ensure the accuracy of the findings, we also removed any subsequent hospitalizations for the same individual if they occurred within a 30-day window from a previous discharge. This exclusion criterion helped to minimize the potential for overlapping treatment periods, providing a clearer picture of the healthcare needs and outcomes for the elderly population in the region.

### 2.3. Health Outcomes and Costs

The annual hospitalization rates per 100,000 individuals were determined by dividing the yearly number of hospitalizations by the size of the resident population over 65 (source: Demo Istat [24]). The length of stay (LOS) was computed as the difference in days between the admission and discharge dates. To calculate the case fatality rate (CFR), the number of in-hospital deaths was divided by the number of patients hospitalized with pneumonia-related diagnoses, expressed as a percentage. In addition, comorbidities were evaluated by reviewing the discharge diagnoses and then calculating the Charlson Comorbidity Index (CCI) [25].

The costs were calculated based on the length of hospital stay, the treatment plan, and the diagnosis-related group (DRG) linked to the discharge diagnoses. Under the DRG-based payment system, each admitted patient falls into a group of similar diagnostic cases, indicating that patients within each group have comparable clinical conditions, are anticipated to require the same level of hospital resources, and result in similar expenses. The expenses for each DRG in Veneto are outlined in the regional tariff schedule [26]. All costs are denoted in euros (EUR).

### 2.4. Statistical Analysis

Categorical variables were represented with frequencies and percentages, while continuous variables were summarized as means, standard deviations (std), medians, minimum–maximum values, and interquartile ranges (IQRs). Pearson’s Chi-squared and Kruskal–Wallis rank sum tests were employed to assess differences in HDR variables between age groups 65–74, 75–84, and over 85 years. Joinpoint regressions [27] were conducted to evaluate the significance of hospitalization rates, CFR, costs, and LOS trends over the years. The results were expressed as Annual Percentage Change (APC) and average APC (AAPC). Confidence intervals (CIs) were calculated as appropriate. Given the large sample size, a *p*-value of <0.001 was considered significant in the bivariate analysis results and a *p*-value of <0.05 was considered significant for Joinpoint results.

All data manipulations, analyses, and visualizations were performed using Python 3.8.18 and R 4.2.2.

### 2.5. Ethics Statement

Hospital Discharge Records (HDRs) were obtained from the administrative databases of the Veneto Region. The disclosure and utilization of such records for educational and scientific purposes do not necessitate approval from ethical committees. On 24 January 2023, the Veneto Region implemented the Code of Conduct for the use of health data for educational and scientific publication purposes (Official Bulletin of the Region, “*Bollettino Ufficiale della Regione*” no. 10), as established by the European Committee (European Regulation 2016/679). This implementation received approval from the Italian Personal Data Protection Authority on 14 January 2021.

Adhering to the current Italian privacy legislation, the publication and utilization of HDR data, along with the processing methods, must occur exclusively in aggregate form, without any reference to patients’ personal information. Prior to the authors gaining access to these data, all personal data that could potentially lead to identification were substituted with anonymous codes, in accordance with current privacy regulations (Legislative Decree no. 196 of 30 June 2003).

## 3. Results

In total, 139,201 pneumonia-related hospitalizations involving elderly individuals from 2007 to 2023 were identified from the Veneto Region HDR database. The majority of patients were male (51.2%) and aged ≥85 years (44.5%). Only 2.0% of discharge records reported a specific diagnosis of pneumococcal pneumonia (ICD-9-CM 481), while for the majority, the etiological agent was unspecified.

Associated meningitis, septicemia, and empyema were reported as the primary diagnoses in 68 (0.05%), 6119 (4.4%), and 219 (0.2%) cases, respectively.

More than half of the patients (57.2%) had at least one comorbidity among their secondary diagnoses. The most frequently detected conditions were cardiovascular diseases (29.4%), including heart failure and cerebrovascular accident, diabetes (11.1%), chronic obstructive pulmonary disease (COPD, 9.9%), and moderate to severe chronic kidney disease (CKD, 6.7%). The estimated median Charlson Comorbidity Index was 4 (IQR 4; 5). Almost all hospitalizations followed an emergency room urgent admission (92.1%). A higher number of pneumonia-associated hospital admissions were recorded in winter (31.1%), confirming the seasonality of pneumonia epidemiology (Appendix A, Figure A1).

The overall in-hospital case fatality rate was 14.9%, the median length of stay was 10 days (IQR 7-15), and the median inpatient direct cost per hospitalization was EUR 3307.1 (IQR EUR 2445.4; EUR 3307.1). The characteristics of the sample and the bivariate analysis by age group are shown in Table 2.

Statistically significant differences (*p*-value < 0.001) between the age groups were found for almost all of the investigated variables. Among hospitalized patients aged ≥85 years, there were more females (58.8% versus 37.0% and 42.5% in the 65–74 and 75–84 years age groups), smaller proportions of pneumococcal pneumonia (1.5% versus 3.4% and 2.1%), meningitis (0.02% versus 0.1% and 0.05%), and empyema diagnosis (0.1% versus 0.4% and 0.2%), a higher CFR (20.6% versus 6.5% and 12.0%), but a slightly lower average cost (EUR 3144.2 versus 3360.3 and 3278.7). No departures in terms of median LOS were observed. The observed prevalence of reported diagnoses for cardiovascular diseases and moderate–severe CKD increased with age, rising from 16.5% to 35.9% and 6.1% to 7.1%, respectively. Conversely, hospitalized individuals aged 65–74 years were more frequently associated with secondary diagnoses of COPD (11.7% versus 11.5% and 7.8% in the 75–84 and over 85 years age groups), diabetes (14.0% versus 12.0% and 9.1%), and cancer, including localized or metastatic tumor, leukemia, or lymphoma (9.5% versus 6.3% and 3.2%).

The overall pneumonia-related hospitalization rate ranged from 437.81 to 902.44 cases per 100,000, with the highest value in 2015 and the lowest in 2021. The data in Figure 1 show that before the pandemic years, hospitalization rates remained stable at approximately 850 cases per 100,000 inhabitants (APC 2007–2018: 0.607, 95% CI −0.45; 2.05). There was a strong decline in 2020–2021 (APC 2018–2021: −19.697, 95% CI −23.33; −13.57), followed by an increase to 624.35 in 2023 (APC 2021–2023: 15.001, 95% CI 3.61; 26.14). Overall, the average APC during the study period was negative (−1.931, 95% CI −2.94; −0.93); however, a significant positive trend was found for specified pneumococcal pneumonia (AAPC +6.820, 95% CI 3.48; 10.37). Figure 2a shows significant differences in pneumonia-related hospitalization rates by age groups: rates in those aged ≥85 years were on average 9.13 and 3.91 times higher than in those aged 65–74 and 75–84 years. The trends by age follow the overall pattern. Figure 2b depicts hospitalization rates related to other co-diagnosed invasive bacterial diseases. Discharges with meningitis or empyema remained stable throughout the study period (AAPC 95% CIs include 0). In contrast, pneumonia-related septicemia hospitalizations experienced an upward trend before the COVID-19 pandemic (APC 2007–2018: 6.752, 95% CI 4.19; 16.47).

The estimated in-hospital case fatality rate trend showed an overall increase (AAPC 1.167, 95% CI 0.31; 1.84), although it was stationary between 13% and 15.5% before the COVID-19 emergency (Figure 3a), and it was not significant when stratified by age. Similarly, the trend of mean costs showed a significant increase, rising from EUR 3062.5 to more than EUR 3301.5 (Figure 3c). Conversely, the length of stay showed a slight decrease (Figure 3b).

## 4. Discussion

This retrospective observational study analyzed trends in pneumonia-related hospitalizations among the elderly population in the Veneto Region from 2007 to 2023, confirming that the health and resource burden posed by pneumonia remains a significant public health concern.

The overall pneumonia-related hospitalization rate in individuals over 65 decreased during the investigated period (AAPC −1.931). However, the trend is stable when only the pre-pandemic years are considered, while a decline can be observed starting from 2020 (AAPC −19.697), followed by an increase in 2023 (APC 15.001). The observed increase in hospitalization rate has not reached the pre-pandemic levels. Therefore, further studies are necessary to understand the evolution in the coming years. Instead, a significant positive trend was observed for pneumococcal pneumonia (AAPC +6.820), although only 2% of discharge forms reported a specific diagnosis. These findings are consistent with those of a previous Italian study by Amodio et al., which reported an increased risk of hospitalization for specified bacterial pneumonia (AAPC +3.33) in adults (≥18 years) from 2010 to 2019 [28]. Additionally, the authors noted a significant positive trend at the national level for unspecified pneumonias. A Spanish study also observed an increase in pneumococcal pneumonia cases among both children and the elderly in the years preceding the COVID-19 pandemic [29].

Notably, we observed a strong decline in the number of admissions related to pneumonia in 2020–2021 (APC 2018–2021: −19.697). While the values for 2023 showed an upward trend, they remained lower than those observed before 2019. The SARS-CoV-2 pandemic could have been an important factor influencing the trend in hospitalizations for bacterial pneumonia. The implementation of individual and collective prevention measures, such as the use of face masks and the physical distancing of people, has generally led to less circulation of respiratory microorganisms. In more recent times, there has been an increase in pneumonias. On the one hand, this increase is probably due to the relaxation of containment measures and, on the other hand, to the continuing use of diagnostic tests for SARS-CoV-2 that allow the exclusion of the diagnosis of COVID-19 [30].

Our study confirms that older age and male sex are associated with an increased risk of hospitalization, as already reported in the literature [31,32]. Among individuals aged ≥85 years, the hospitalization rate was, on average, 9.13 times higher than that among those aged 65–74 years and 3.91 times higher than that among those aged 75–84 years. Our estimates are similar to those reported in a recent meta-analysis. Shi et al. [31] estimated based on the results of 49 studies that the hospitalization rate for individuals over 65 was 5.1 times higher than that for those aged 65–74 and 2.1 times higher than that for those aged 75–84 in industrialized countries. In developing countries, the corresponding rates were 9.3 and 3.12 times higher, respectively. Among elderly males, the incidence of hospitalization was found to be 1.3 times that of females in industrialized countries (from seven articles), although no comparable data were available for developing countries.

Cardiovascular disease (29.4%), diabetes (11.1%), and COPD (9.9%) were the most commonly reported comorbidities. Although not investigated in our study, several articles have highlighted that these conditions are linked to an increased risk of hospitalization, clinical deterioration, or mortality following pneumonia [9,28,33,34]. Interestingly, Corrales-Medina et al. reported that hospitalization for pneumonia was associated with an elevated short- and long-term risk of cardiovascular disease, with a hazard ratio ranging from 2.10 to 4.07 in the first year, suggesting that infection may be a risk factor for cardiovascular disease [35]. It is also worth noting that pneumonia has been found to be associated with various socio-demographic and lifestyle factors, such as obesity and smoking. Behavioral and environmental factors can increase the risk of hospitalization or mortality [34,36,37,38]. Unfortunately, we were unable to analyze these factors in our study because they were not available in the HDRs.

In adults, and especially among the elderly, mortality from pneumonia remains quite high. Overall, we observed a case fatality rate of 14.9%, which increased progressively with age, reaching 20.6% in individuals over 85. Our findings are consistent with those of the national study by Amodio et al. [28], which reported an in-hospital death rate of 13.0% in patients aged ≥18. Analyzing the temporal evolution, we estimated a slight increase in the CFR (AAPC +1.167), although this increase was not significant when stratified by age group. Other authors have also observed an increase in mortality among adults and the elderly, while a general decrease in mortality has been reported among children, particularly those under 5 years old [17,28,39,40]. This could highlight the impact of PCV on younger people [17]; however, there is limited evidence on the elderly.

The increase in healthcare burden has also been followed by an increase in mean inpatient spending, although the median cost associated with the DRG for pneumonia has remained unchanged over the years (EUR 3307). In the Veneto Region, we observed an increase in overall healthcare expenditure, although the average length of stay decreased. Additionally, we found that hospitalization costs decrease with age, a pattern that has also been observed in Dutch and English population studies [41,42]. Specifically, Campling et al. reported that in adult patients ≥18, costs increased up to 75 years of age, even though the length of stay continued to rise beyond this age [42]. The authors also noted that the commonly reported higher costs associated with a diagnosis of pneumococcal pneumonia (as observed in our population; see Appendix A, Table A1) may be biased upwards if the causative organism is more likely to be identified in cases of severe, complicated, or antibiotic-resistant disease. Indeed, in our cohort, there were significant differences in the prevalence of meningitis, septicemia, and empyema among pneumonias with a specified diagnosis of pneumococcus (Appendix A, Table A1).

Our study has several limitations. The first is the reliance on HDRs from the Veneto Region, as these data are primarily used as an administrative tool and only secondarily used for healthcare purposes. Accessing complete medical histories was not feasible within the scope of this research. As a result, our classification depends on the diagnosis codes recorded in the HDRs, which are still coded using the ICD-9-CM system in the Veneto Region. Moreover, HDRs may introduce quantitative distortion due to the coding practices and reverse reporting bias. While they demonstrate a high specificity in identifying severe cases, they have only limited sensitivity. As a result, pneumonia cases with other co-diagnoses may have been neglected. In addition to possible miscoding, we emphasize that our study may also be susceptible to underestimation of cases of bacterial pneumonia due to other etiologic agents or co-infection with other etiologic agents.

Furthermore, HDR data do not contain any information about the vaccination status of the patient. Finally, our sample could limit the generalizability of the results to other populations. Despite these constraints, the present work covers a long period, allowing for the collection of a large sample of data.

## 5. Conclusions

This study underscores the significant burden that pneumonia poses on individuals aged over 65, with profound implications for public health due to its high case fatality rate and the associated healthcare costs. A portion of these hospitalizations might be preventable through the implementation of targeted preventive measures. In the Veneto Region, the available data (2016–2022) show that the vaccination coverage of 65-year-old individuals, in recent years and based on the age cohort of the call for vaccination, fluctuates between 55.1% and 57.4% of the vaccine-eligible population for pneumococcal disease according to regional guidelines [43]. Since vaccination prevents serious disease, we can see how an increase in pneumococcal vaccination coverage is crucial in reducing hospitalization and related healthcare expenditures. Pneumococcal vaccination is the best strategy to combat the various pneumococcal-related diseases. Therefore, it is important to inform the population of the advisability of the vaccine. At the European level, striving for greater uniformity of vaccine offerings and coverage will be beneficial [11].

## Figures and Tables

**Figure 1 diseases-12-00254-f001:**
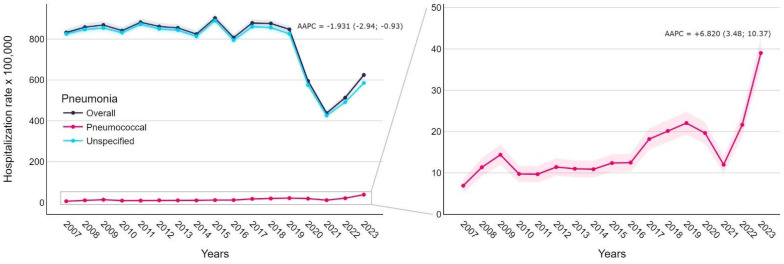
Pneumonia-related hospitalization rate (per 100,000 inhabitants) trends with average Annual Percentage Change (AAPC) and 95% confidence intervals.

**Figure 2 diseases-12-00254-f002:**
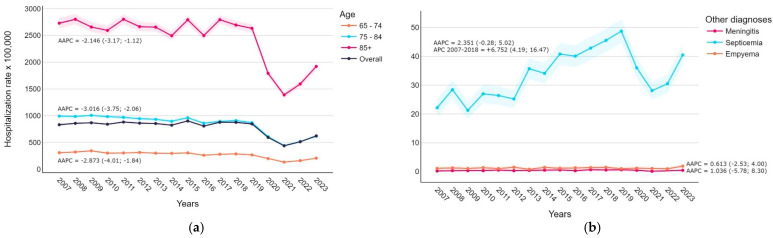
Hospitalization rate (per 100,000 inhabitants) trends with average Annual Percentage Change (AAPC) and 95% confidence intervals: (**a**) pneumonia-related admissions by age group; (**b**) other co-diagnosed invasive bacterial diseases.

**Figure 3 diseases-12-00254-f003:**
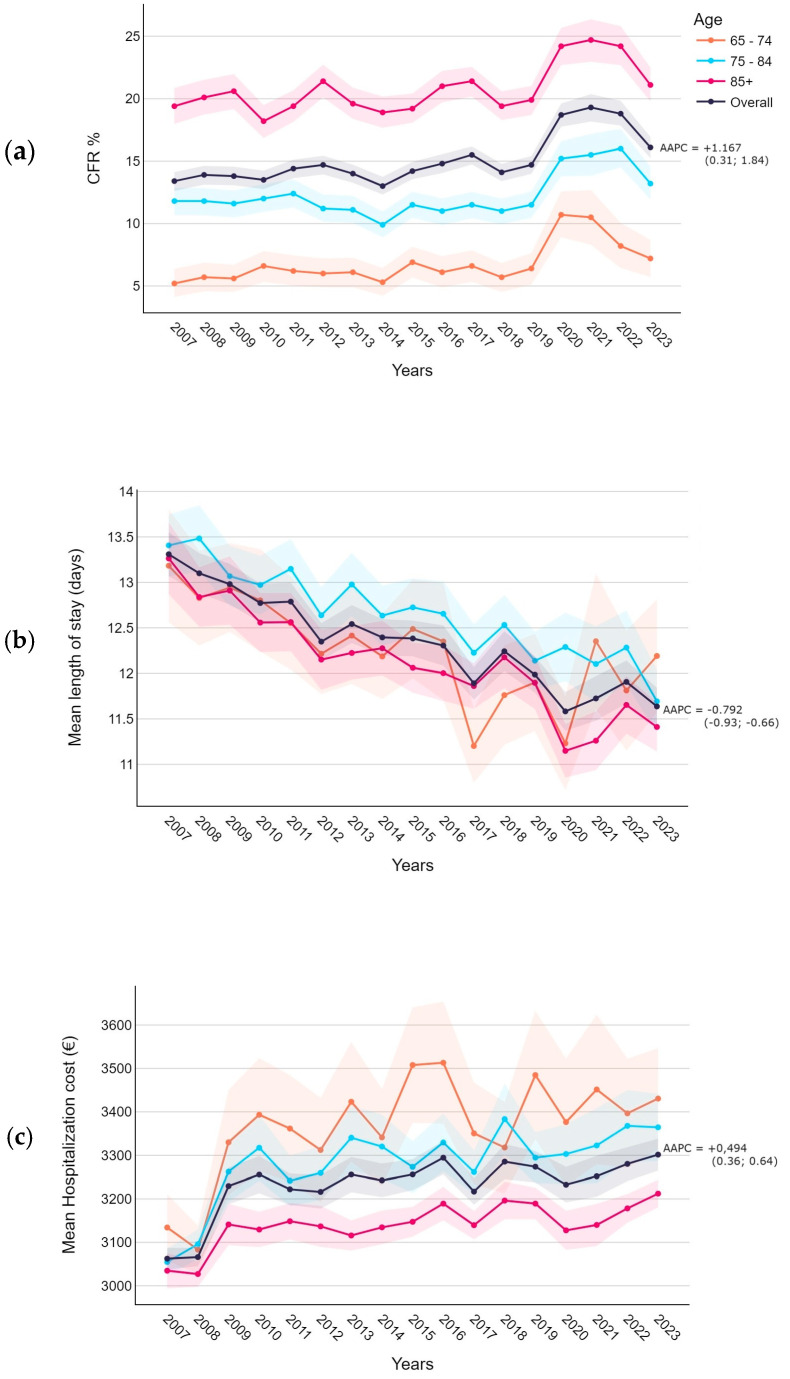
Case fatality rate (**a**), length of stay (**b**), and hospitalization cost (**c**) trends with average Annual Percentage Change (AAPC) and 95% confidence intervals.

**Table 1 diseases-12-00254-t001:** Diagnostic codes used for hospital discharge record selection.

Pneumonia-Related Hospitalization Discharge Diagnoses Group	International Classification of Diseases 9 Clinical Modification (ICD-9-CM) Codes	
	First-Listed Discharge Diagnosis	Another Discharge Diagnosis
**Pneumococcal pneumonia** **[*S. pneumoniae* pneumonia]**	481	
**Unspecified pneumonia [pneumonia** **without a causative organism identified]**	485–487; 482.9	
**Meningitis**	321, 013.0, 003.21, 036.0, 036.1, 047, 047.0, 047.1, 047.8, 047.9, 049.1, 053.0, 054.72, 072.1, 091.81, 094.2, 098.82, 100.81, 112.83, 114.2, 115.01, 115.11, 115.91, 130.0, 320, 320.0, 320.1, 320.2, 320.3, 320.7, 320.81, 320.82, 320.89, 320.8, 320.9, 322, 322.0, 322.9	Plus 481; 485–487; 482.9
**Septicemia**	038.1, 038.4, 003.1, 020.2, 022.3, 031.2, 036.2, 038, 038.0, 038.2, 038.3, 038.8, 038.9, 054.5, 790.7	Plus 481; 485–487; 482.9
**Empyema**	510	Plus 481; 485–487; 482.9

**Table 2 diseases-12-00254-t002:** Characteristics of pneumonia-associated hospitalizations by age group.

Variable		Age Hospitalization			Total (N = 139,201)	*p*-Value
		65–74 (N = 24114)	75–84 (N = 53,148)	85+ (N = 61,939)		
**Age hospitalization**	Min/Max	65.0/74.0	75.0/84.0	85.0/121.0	65.0/121.0	<0.0001
	Med [IQR]	71.0 [68.0;73.0]	80.0 [78.0;83.0]	89.0 [87.0;92.0]	83.0 [77.0;89.0]	
	Mean (std)	70.2 (2.8)	80.0 (2.8)	89.8 (3.8)	82.7 (7.9)	
**Sex**	Female	8924 (37.0%)	22,579 (42.5%)	36,439 (58.8%)	67,942 (48.8%)	<0.0001
	Male	15,190 (63.0%)	30,569 (57.5%)	25,500 (41.2%)	71,259 (51.2%)	
**Pneumonia**	Pneumococcal	808 (3.4%)	1102 (2.1%)	933 (1.5%)	2843 (2.0%)	<0.0001
	Unspecified	23,306 (96.6%)	52,046 (97.9%)	61,006 (98.5%)	136,358 (98.0%)	
**Other diagnosed invasive diseases**	Meningitis	32 (0.1%)	24 (0.05%)	12 (0.02%)	68 (0.05%)	<0.0001
	Septicemia	1084 (4.5%)	2388 (4.5%)	2647 (4.3%)	6119 (4.4%)	0.1375
	Empyema	94 (0.4%)	86 (0.2%)	39 (0.1%)	219 (0.2%)	<0.0001
**Admission season**	Winter	7591 (31.5%)	16,568 (31.2%)	19,090 (30.8%)	43,249 (31.1%)	0.0020
	Spring	6012 (24.9%)	12,868 (24.2%)	14,908 (24.1%)	33,788 (24.3%)	
	Summer	4933 (20.5%)	11,391 (21.4%)	13,331 (21.5%)	29,655 (21.3%)	
	Autumn	5578 (23.1%)	12,321 (23.2%)	14,610 (23.6%)	32,509 (23.4%)	
**Comorbidities**	Cardiovascular diseases	3989 (16.5%)	14,671 (27.6%)	22,244 (35.9%)	40,904 (29.4%)	<0.0001
	COPD	2816 (11.7%)	6117 (11.5%)	4839 (7.8%)	13,772 (9.9%)	<0.0001
	Moderate–severe CKD	1471 (6.1%)	3397 (6.4%)	4393 (7.1%)	9261 (6.7%)	<0.0001
	Diabetes	3381 (14.0%)	6368 (12.0%)	5662 (9.1%)	15,411 (11.1%)	<0.0001
	Neoplasms	2296 (9.5%)	3326 (6.3%)	1962 (3.2%)	7584 (5.4%)	<0.0001
	Other comorbidities	2156 (8.9%)	6634 (12.5%)	10,460 (16.9%)	19,250 (13.8%)	<0.0001
**Charlson Comorbidity Index**	Min/Max	2.0/17.0	3.0/17.0	4.0/18.0	2.0/18.0	<0.0001
	Med [IQR]	3.0 [3.0;4.0]	4.0 [4.0;5.0]	5.0 [4.0;5.0]	4.0 [4.0;5.0]	
	Mean (std)	3.7 (1.7)	4.6 (1.5)	5.0 (1.3)	4.6 (1.5)	
**Deceased (CFR%)**		1574 (6.5%)	6377 (12.0%)	12,755 (20.6%)	20,706 (14.9%)	<0.0001
**Length of stay (days)**	Min/Max	0/288.0	0/194.0	0/501.0	0/501.0	<0.0001
	Med [IQR]	10.0 [7.0;15.0]	10.0 [7.0;15.0]	10.0 [7.0;15.0]	10.0 [7.0;15.0]	
	Mean (std)	12.3 (10.5)	12.7 (9.8)	12.1 (9.7)	12.4 (9.9)	
**Hospitalization cost (EUR)**	Min/Max	200.0/4.3 × 10^4^	200.0/3.8 × 10^4^	200.0/6.8 × 10^4^	200.0/6.8 × 10^4^	<0.0001
	Med [IQR]	3307.1 [2445.4;3307.1]	3307.1 [2445.4;3307.1]	3307.1 [2445.4;3307.1]	3307.1 [2445.4;3307.1]	
	Mean (std)	3360.3 (2370.9)	3278.7 (1885.3)	3144.2 (1217.3)	3232.9 (1730.7)	
	N (NA)	23,981 (133)	52,934 (214)	61,795 (144)	138,710 (491)	
**Hospitalization cost per day (EUR)**	Min/Max	68.3/6682.9	5.7/1.3 × 10^4^	80.9/9924.2	5.7/1.3 × 10^4^	<0.0001
	Med [IQR]	300.6 [206.7;416.2]	300.6 [200.0;413.4]	300.6 [200.0;413.4]	300.6 [200.0;413.4]	
	Mean (std)	379.8 (339.3)	362.6 (314.2)	376.1 (309.1)	371.6 (316.5)	
	N (NA)	23,981 (133)	52,934 (214)	61,795 (144)	138,710 (491)	
**DRG type**	Surgical	406 (1.7%)	398 (0.8%)	147 (0.2%)	951 (0.7%)	<0.0001
	Medical	23,575 (98.3%)	52,536 (99.2%)	61,648 (99.8%)	137,759 (99.3%)	
	NA	133	214	144	491	
**Emergency admission**		21,835 (90.5%)	48,738 (91.7%)	57,635 (93.1%)	128,208 (92.1%)	<0.0001

Legend: NA: not available; IQR: interquartile range; std: standard deviation; CFR: case fatality rate; COPD: chronic obstructive pulmonary disease; CKD: chronic kidney disease; and DRG: diagnosis-related group.

## Data Availability

The data that support the findings of this study are available upon request from the corresponding author.

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
