# Peer review of "Pneumonia-Related Hospitalizations among the Elderly: A Retrospective Study in Northeast Italy"

_diseases, 2024, doi:10.3390/diseases12100254_

Round 1

Reviewer 1 Report

Comments and Suggestions for Authors

This is an interesting retrosepctive study that give us information about pneumonia in the elderly in the Veneto Region from 2007 to 2023. The authors reported that during 2007 to 2023, there were 139,201 hospitalizations for pneumonia, and discharge mortality rates ranged from 13% to 19.3%,
significantly higher in those over 85 (20.6% compared to 6.5%, 12.0% in the 65-74, 75-84 age groups respectively).

My major concern is related to the inclusion and exclusion criteria,and the diagnosis of pneumonia. There is not clear explanation for why they authors decide to exclude viral pneumonia.

Were cases of aspiration pneumonia included?

I suggest removing the term HCAP because is no longer because the HCAP criteria performed poorly as a predictive tool to identify MDR pneumonia or pathogens not covered by treatment for CAP.

Also, Il will be helpful to add a limitation about the diagnosis of pneumonia

Author Response

Comment: My major concern is related to the inclusion and exclusion criteria, and the diagnosis of pneumonia. There is not clear explanation for why they authors decide to exclude viral pneumonia.

Response: Thank you for raising this concern. We carefully considered this thoughtful observation and ultimately decided to remain aligned with our original study focus. The primary aim of our study was to provide epidemiological estimates specifically for bacterial pneumonia, with a focus on pneumococcus-related pneumonia in the elderly population. We chose to exclude viral pneumonias because, as supported by the literature, bacterial infections are the most common cause of pneumonia in adults.

Comment: Were cases of aspiration pneumonia included?

Response: Aspiration pneumonia were not included because we could not discriminate whether those cases are related to bacterial pneumoniae, in particular pneumococcus-related pneumonia, from hospital discharge records. We tried to retrieve the cases with “aspiration pneumonia” from our sample using the following ICD-9 codes:

  • 997.32: “Postprocedural aspiration pneumonia”
  • 507.0: “Pneumonitis due to inhalation of food or vomitus”

We found only 106 (<0.08%) patients with a diagnosis of “Pneumonitis due to inhalation of food or vomitus” (507.0).

Comment: I suggest removing the term HCAP because is no longer because the HCAP criteria performed poorly as a predictive tool to identify MDR pneumonia or pathogens not covered by treatment for CAP.

Response: Thank you, we corrected as suggested.

Comment: Also, Il will be helpful to add a limitation about the diagnosis of pneumonia

Response: We improved the limitation paragraph according to your suggestion.

All changes have been highlighted in the revised manuscript for ease of reference. We would like to thank you for the time and effort dedicated to reviewing our manuscript. We sincerely appreciated the feedback, which has allowed us to improve the quality and clarity of our work. 

Reviewer 2 Report

Comments and Suggestions for Authors

The burden of pneumonia, especially in the elderly population, is considerable. While there are older studies quantifying the burden of disease, there are few studies from the COVID-19 pandemic and the period towards the end of the pandemic.

A study from the north-eastern part of Italy contributes to the insight into the effect of the COVID-19 pandemic on pneumonia hospitalizations and mortality during and immediately after the pandemic, and provides data on the economic burden caused by pneumonia hospitalizations.

Two minor remarks:

Abstract – Line 22 and 25- DRG and AAPC – explain the abbreviation when used for the first time.

Figure 3 a an b – time line is missing on the abscissa axis of the graph. Figure 3c is missing.

Author Response

Comment: Abstract – Line 22 and 25- DRG and AAPC – explain the abbreviation when used for the first time.

Response: Done. Thank you.

Comment: Figure 3 a an b – time line is missing on the abscissa axis of the graph. Figure 3c is missing.

Response: We fixed Figure 3, thank you.

All changes have been highlighted in the revised manuscript for ease of reference. We would like to thank you for the time and effort dedicated to reviewing our manuscript. We sincerely appreciated the feedback, which has allowed us to improve the quality and clarity of our work. 

Round 2

Reviewer 1 Report

Comments and Suggestions for Authors

The authors have responded to suggestions on the article, it is an interesting article. Congratulations to the authors for the work.